

# A 2D model for simulating heterogeneous mass and energy fluxes through melting snowpacks

Nicolas R. Leroux[1] and John W. Pomeroy[1]

[1]Centre for Hydrology, University of Saskatchewan, Saskatoon, Canada

*Correspondence to*: Nicolas R. Leroux (nicolas.leroux@usask.ca)

**Abstract.** Accurate estimation of the water flux through melting snowpacks is of primary importance for runoff prediction. Lateral flows and preferential flow pathways in porous media flow have proven critical for improving soil and groundwater flow models, but though many physically-based layered snowmelt models have been developed, only 1D matrix flow over level ground is currently accounted for in snow models. Snowmelt models that include these processes may improve snowmelt

discharge timing and contributing area calculations in hydrological models. A two-dimensional snow model (SMPP – Snowmelt Model with Preferential flow Paths) is presented that simulates heat and water flows through both snowpack matrix and preferential flow paths, as well as snowmelt and refreezing of meltwater. The model assumes thermodynamic equilibrium between solid and liquid phases and uses the latest improvements made in snow science to estimate snow hydraulic and thermal properties. A finite volume method is applied to solve for the 2D heat and water equations. The use of a water entry pressure

for dry snow combined with consideration of the impact of heterogeneities in surface fluxes and internal snow properties – density, grain size and layer thickness – allowed calculation of the formation of preferential flow paths in the snowpack. The simulation of water flow through preferential flow paths resulted in liquid water reaching the base of the snowpack earlier than for a homogeneous wetting front. Moreover, the preferential flow paths in the model increased the exchange of energy between the snow surface and the internal snowpack, resulting in faster warming of the snowpack. A sensitivity analysis, conducted on

the snow internal properties showed that initial conditions such as density and temperature, should be carefully measured in the field to accurately estimate liquid water percolating through the snowpack. Furthermore, two empirical coefficients used in the water flow equation were showed to greatly impact model outputs. This heterogeneous flow model is an important tool to help understand snowmelt flow processes in complex and level terrains and to demonstrate how uncertainty in snowmelt-derived runoff calculations might be reduced.

Keywords: snowmelt, preferential flow paths, two-dimensional snow model, heat and mass transfers

# 1 Introduction

To accurately predict the timing and magnitude of snowmelt runoff from deep cold snowpacks, water flow percolation within snow must be understood (Gray and Male, 1975; Wankiewicz, 1979). Liquid water flow within the snowpack is influenced by



its internal properties. Deeper, colder snowpacks have slower flow rates; this lag and attenuation in timing of meltwater delivery to the soil surface makes the process important for runoff and streamflow generation in mountains. The water flow through snow is a complex physical process that can be divided into two flows – matric flow and preferential flow (e.g. Marsh and Woo, 1984a; Marsh, 1991; Waldner et al., 2004).

5      Many theories have arisen to describe the matric flow through snow. The gravitational vertical flow percolation within a homogeneous, isothermal snowpack was first described by Colbeck (1972) and Colbeck and Davidson (1973), the refreezing of liquid water percolating a subfreezing, layered snowpack with phase change was modelled by Illangasekare et al. (1990), Pfeffer et al., (1990) and Daanen and Nieber (2009) and the influence of capillary forces on the water flow was represented by implementing Richards equation in a snow model (e.g. Jordan, 1995; Hirashima et al., 2010; Wever et al., 2014).

10      Several numerical snowmelt models with differing levels of complexity have been developed in the past decades. Tseng et al. (1994) developed a complex two-dimensional snow model based on the theory of Illangasekare et al. (1990), but this theory has not been validated against *in-situ* data and preferential flow paths were not represented. The Flow Impeding Neutral or Accelerating (FINA) by Wankiewicz (1979) was able to simulate the acceleration or impeding of water flow at the interface of two snow layers depending on the gravity flow pressures of these layers. However, no field data was available at the time 15 to test and validate his theory. Marsh and Woo (1985) created a one-dimensional model that accounted for the mass flow through different preferential flow pathways; however, this theory does not include lateral flows, the delay of water flow due to ice layers, and it assumed that each preferential flow path (PFP) extends over the complete depth of the snowpack. In addition, no operational snow model in avalanche forecasting such as Crocus (Brun et al., 1989, 1992) and SNOWPACK (Bartelt and Lehning, 2002; Bartelt et al., 2002; Lehning et al., 2002), snow hydrological models such as SNTHERM (Jordan, 20 1991) and Snobal (Marks et al., 1999) or land surface scheme is able to predict lateral flows, meltwater flows on slopes, the formation of PFPs and ice layers, and their effects on the water flow through snow; this results in inaccuracy in snowpack and atmospheric energetics as well as in the prediction of catchment discharge and meltwater delivery to soil (Pomeroy et al., 1998).

     The formation of PFPs in soil has been studied for decades. Hill and Parlange (1972) demonstrated that PFP in soils form 25 after liquid water ponds at the interface of fine to coarse structured layers and when the wetting front does not stay horizontal flat but becomes unstable. This ponding of water was later shown to also occur in snow (Waldner et al., 2004). Hillel and Baker (1988) emphasized on the importance of water-entry suction on the ponding of percolating liquid water. Water-entry suction was defined as "the maximum suction that will allow water to enter an initially dry snow", which is defined by the smallest pores in a layer. Therefore, before liquid water penetrates an initially dry layer, it accumulates until the suction at the 30 interface drops below the water-entry suction threshold. However, liquid water will first penetrate the dry sublayer at randomly distributed locations caused by spatial heterogeneities at the interface. This ponding of liquid water induced by the water-entry suction of a dry porous medium was proved to occur and measured for dry snow (Katsushima et al., 2013).

     The modelling of flow finger formation in initially air-dry and repellent sandy soils was first undertaken by Nieber (1996) and Ritsema et al. (1998). Both numerical studies implemented a water-entry capillary pressure and were initialized with





unstable wetting fronts. Ritsema et al. (1998) also showed the persistence and recurrence of water flow through PFP with their model.

Based on soil studies and the works of Waldner et al. (2004) and Katsushima et al. (2013), Hirashima et al. (2014) developed a multi-dimensional model to reproduce preferential flows in snow. They introduced a new water-entry capillary pressure for dry snow, as well as snow grain and density heterogeneities to trigger the formation of PFP. Hirashima's model includes the latest improvements made to estimate snow hydraulic properties, such as the formulation of Calonne et al. (2012) for estimating the hydraulic conductivity of saturated snow and the empirical model of Yamaguchi et al. (2012) to approximate the water entry pressure in wet snow. However, their model was limited to small artificial and isothermal snow samples, neglecting the snow layering system, melting at the surface, and refreezing of liquid water.

In this paper, a novel numerical snowmelt model, SMPP (Snowmelt Model with Preferential flow Paths) is presented. This study aims to show the different numerical methods this two-dimensional model uses to trigger the formation of preferential flow paths during melt of a dry, subfreezing and layered snowpack. Finally, the results of a sensitivity analysis carried on the model inputs and parameters are presented in order to identify the most important boundary and initial conditions and snowpack properties that influence flow through snow.

## 2 Mathematical model

### 2.1 Snow ablation and melt

A melting snow surface is a moving boundary at which heat transfer and phase change occur simultaneously. To estimate the heat transfer and phase change at this boundary, Stefan condition is solved (Eq. 1) (e.g. Tseng et al., 1994).

$$Q_n = -\lambda \frac{\partial T}{\partial z}(z = S) + L_f \rho_s V_n \tag{1}$$

where,

$$Q_n = -\lambda \frac{\partial T}{\partial z}(z = S) \quad \text{if} \quad T_s < 0^\circ C$$

$$Q_n = L_f \rho_s V_n \quad \text{if} \quad T_s = 0^\circ C$$

and $Q_n$ is the heat flux at the surface [W/m$^2$], $\lambda$ is the thermal conductivity [W/(K.m)], $\partial T / \partial z$ the vertical temperature gradient at the surface [K/m], $L_f$ the latent heat of fusion of ice [J/kg], $\rho_s$ the snow density [kg/m$^3$], $V_n$ the velocity of the melting snow surface [m/s], and $T_s$ is the snow surface temperature [$^\circ$C].

The infiltration rate (Eq. 2) can be estimated from the vertical velocity of the melting snow surface $V_n$.

$$Q_{inf} = V_n(\frac{\rho_s}{\rho_w} + \theta) \tag{2}$$

where $Q_{inf}$ is the infiltration rate at the snow surface [m/s], $\rho_w$ the density of water [kg/m$^3$] and $\theta$ the volumetric liquid water content within the melting volume.



## 2.2 Water flow

The mass flow between each snow layer is estimated by solving the two-dimensional Richards equation:

$$\frac{\partial \theta}{\partial t} + \nabla \vec{q} = S_s \tag{3}$$

where q is the macroscopic flow velocity [m/s] (Eq. 4) and $S_s$ is a mass sink term caused by refreezing of liquid melt water [s$^{-1}$].

The macroscopic flow velocity in an unsaturated medium is commonly estimated from Darcy's law under the condition that the flow is laminar.

$$\vec{q} = K(\theta)\, \vec{\nabla}(\Psi(\theta) + z) \tag{4}$$

where $K(\theta)$ is the unsaturated hydraulic conductivity [m/s], $\Psi(\theta)$ is the matric head [m], and z the height from a stratum [m]. For unsaturated porous media, both K and $\Psi$ are functions of the water content.

In snow science, studies have been conducted to establish relationships between snow hydraulic properties and water content. Calonne et al. (2012) developed a relationship between saturated hydraulic conductivity ($K_s$), dry snow density, and optical grain size (Eq. 5) by solving Stokes equation in three-dimensional tomographic images of snow samples. Knowing the saturated hydraulic conductivity, the unsaturated hydraulic conductivity can be estimated (e.g. Colbeck and Davidson, 1973; Mualem, 1976).

$$K_s = (3 \pm 0.3)\frac{\rho_w g}{\mu_w} r_{opt}^2 \exp(-(0.013 \pm 0.0003)\rho_{ds}) \tag{5}$$

with g the gravitational constant [m/s$^2$], $\mu_w$ the dynamic viscosity of water [Pa.s], $r_{opt}$ the optical grain radius [m], which can be related to classical grain size, sphericity and dentricity (Vionnet et al., 2012), and $\rho_{ds}$ the dry snow density [kg/m$^3$]. A sensitivity analysis on the two empirical constants ($3 \pm 0.3$ and $0.013 \pm 0.0003$) is carried in Sect. 5.20.

The water retention curve (WRC) is the relationship between matric head and liquid water content ($\Psi(\theta)$). Analogous to flow through unsaturated soil, the snow WRC has hysteretic behaviour (Wankiewicz, 1979; Adachi et al., 2012). Yamaguchi et al. (2012) developed a WRC for snow based on the van Genuchten model (Eq. 6). Through laboratory experiments, they established empirical equations to link the parameters $\alpha$ and n (cf. Eq. 6) to dry snow density and grain size (Eq. 7). This parameterisation was found to provide better results than the previous formulation of Yamaguchi et al. (2010) (Wever et al., 2015). However, this WRC was developed only for drying snow, i.e. the snow was initially wet and liquid water was draining from it by gravity.

$$S_e = (1 - |\alpha\Psi|^n)^{-m} \tag{6}$$

where $S_e$ is the effective saturation ($S_e = (\theta - \Phi)/(\theta_i - \Phi)$, with $\Phi$ the snow porosity and $\theta_i$ the irreducible water content), and $\alpha$, n, and m are parameters (Eq. 7), with m chosen as $m = 1 - 1/n$.

$$\alpha = 4.4 \times 10^6 \left(2\frac{\rho_{ds}}{r_c}\right)^{-0.98} \tag{7}$$

$$n = 1 + 2.7 \times 10^{-3} \left(2\frac{\rho_{ds}}{r_c}\right)^{0.61}$$





where $r_c$ is the classical grain radius [m].

However, for wetting snow, i.e. snow that is initially dry and into which liquid water infiltrates, the model of Yamaguchi et al. (2012) is not appropriate. Therefore, in the two-dimensional model presented in this paper, a new value of water entry pressure (Eq. 8) is used when the snow is dry - water content is below the irreducible water content level. This pressure,
measured by Katsushima et al. (2013), depends solely on grain size. However, this watere entry pressure can create a pressure gradient from dry to wet snow layers, which would result in a flux of water from dry to wet snow. To prevent this from happening, a condition is put on Darcy's flux: the flow of water occurs from a wet layer to a dry layer only when Darcy's flux (Eq. 4) is positive, i.e. $\frac{\partial \psi}{\partial z} > 1$ . The snow WRC developed by Yamaguchi et al. (2012) is used to estimate the matric head when the snow is wet - water content is greater than the irreducible water content. If the snow is saturated, an air entry pressure
is computed to estimate the water pressure in the pores following de Rooji and Cho (1999).

$$|\Psi_{we}| = 0.0437 \left(\frac{2}{r_c}\right) + 0.01074 \tag{8}$$

where $\Psi_{we}$ is the water entry pressure [m] and the classical grain radius $r_c$  is in [mm].

The implementation of a water entry pressure for dry snow in a numerical snow model was found to allow more liquid water to accumulate at the interface wet-dry snow layers than using Yamaguchi's formulation for both dry and wet snow.

**2.3 Refreezing of liquid water**

In a wet subfreezing snowpack, heat and momentum transfers occur between the liquid and solid phases. Illangasekare et al. (1990) developed a theory to describe the refreezing of meltwater in a subfreezing snowpack. They expressed the maximum mass of liquid water per unit volume of snow ($m_{max}$) that must freeze to raise the snow temperature to zero, i.e. to increase the snow cold content to zero (Eq. 9).

$$L_f m_{max} = -\rho_s C_{p,i} T \tag{9}$$

where T is the temperature of snow layer [K] and $C_{p,i}$ is the specific heat capacity of ice [J/(kg.K)].

However, the real mass of liquid water per unit volume of snow that refreezes during a numerical time step ($m_f$), limited by the available liquid water content in the snow layer, is always less than or equal to $m_{max}$. The new snow layer temperature at the end of a numerical time step $\Delta t$ can be estimated:

$$T^{t+\Delta t} = \frac{\rho_s^t C_{pi} T^t + m_f L_f}{\rho_s^{t+\Delta t} C_{p,i}} \tag{10}$$

At the end of the same time step, snow porosity ($\phi$), effective water saturation ($S_e$), and snow density ($\rho_s$) are updated:

$$\Phi^{t+\Delta t} = \Phi^t + \frac{m_f}{\rho_i} \tag{11}$$

$$S_e^{t+\Delta t} = \frac{\theta^t - m_f/\rho_w}{\Phi^{t+\Delta t}}$$

$$\rho_s^{t+\Delta t} = \rho_s + m_f$$



## 2.4 Heat transfers

To simulate heat transfers in the snowpack, the two-dimensional heat conduction equation is solved following Albert and McGilvary (1992).

$$\left(\rho C_p\right)_s \frac{\partial T}{\partial t} = \frac{\partial}{\partial x_k}\left(\lambda \frac{\partial T}{\partial x_k}\right) \text{ with k=1,2 representing the two spatial directions} \tag{12}$$

such that $\left(\rho C_p\right)_s = \left(\rho_a \theta_a C_{p,a}\right) + \left(\rho_w \theta_w C_{p,w}\right) + \left(\rho_i \theta_i C_{p,i}\right)$

where T is the temperature of a snow layer [K], $\rho$ the density [kg/m$^3$], $C_p$ the specific heat capacity [J/(kg.K)], and $\theta$ the fractional volumetric of each component. The subscripts a, w, and i represent each component of the snowpack: air, water, and ice.

Calonne et al. (2011) conducted three-dimensional numerical computations of snow conductivity through the air and ice

phases. They developed an empirical relationship between thermal conductivity and dry snow density:

$$\lambda = 2.5 \times 10^{-6} \rho_{ds}^2 - 1.23 \times 10^{-4} \rho_{ds} + 0.024 \tag{13}$$

## 3 Numerical model design

A two-dimensional numerical snow model based on the snow physics presented in the previous section was developed to solve for the heat and mass fluxes within a two-dimensional heterogeneous, layered, subfreezing snowpack. To solve the

partial differential equations (PDEs), an explicit finite-volume scheme was used over a quadrilateral structured and unstructured mesh. The size of the mesh is obtained from initial simulations to determine the optimum grid size that allows convergence of the PDEs. This numerical method considers each numerical cell as a control volume, in which the conservation equations are solved. This approach is commonly applied in computational fluid dynamics models as it is inherently conservative. To assure model stability, an adaptive time step is computed so that the Courant-Friedrichs-Lewy conditions for

the two-dimensional heat conduction equation and Richards equation are met (Haverkamp et al., 1977; El-Kadi and Ling, 1993).

### 3.1 Boundary and initial conditions

Neumann boundary conditions were applied at the upper and left-hand boundaries for the mass and heat equations. A constant heat flux ($Q_n$ in Eq. 1) was applied as upper boundary condition for the heat equation. This flux was then used to estimate the

infiltration rate utilized as upper boundary condition for the mass flow equation. The left-hand boundary condition was a no-flow boundary, whereas the lower and right-hand boundaries were set as free drainage boundary conditions.

The snowpack and its properties were initialized before running the model. These data included the snowpack slope angle, the snowpack layering system and the mean layer properties - porosity, water content, grain size, and temperature.





## 3.2 Model assumptions

Water and energy flows within a layered, heterogeneous, subfreezing snowpack are complex physical processes. Therefore, due to the lack of complete understanding of the physics of these processes, it is necessary to make assumptions while developing a numerical snow model. The assumptions made in this model also indicate the current knowledge and how this limits modelling of water flow through melting snow. These assumptions were:

- Changes in grain size due to water vapour gradients (kinematic and equilibrium growth metamorphisms) and presence of liquid water during melt (wet snow metamorphism) were not simulated during the water flow event.
- The irreducible water content was assumed constant for the whole snowpack and did not depend on snow properties.
- Thermal convection, condensation, and sublimation within the snowpack were not simulated.
- Heat conduction dominated the heat transfers.
- Thermodynamic equilibrium between the solid and liquid phases was assumed.
- Freezing point depression effects on the snow grains from pore pressures were neglected.
- The water entry pressure for dry snow was solely a function of snow grain size.
- Temperature, density, and water content were computed at the centre point of each numerical cell and were assumed homogeneous within the cell.

## 3.3 Model verification

The simulation of one-dimensional water flow through unsaturated porous medium using Richards equation with SMRFF was compared to that from the physically detailed soil model Hydrus-1D (Simunek et al., 2008). The flow through a 1 m deep unsaturated sand column was simulated. The soil hydraulic parameters were chosen from the soil catalogue offered with the model Hydrus. The water content was initialized at 0.05 in the whole system, a free drainage boundary condition was chosen for the lower boundary condition and saturation was assumed at the upper boundary. The simulations were run until steady state. Figure 1a shows the outputs from SMRFF against the outputs from Hydrus 1D at three different times. The water flow simulation of the snow model agreed with the 1D simulation from Hydrus suggesting SMRFF will be adequate for water flow simulations through snow using Richards equation.

The 1D heat conduction simulation was validated against the computational fluid dynamics (CFD) model OpenFOAM, using the solver "laplacianFoam" with an explicit finite volume scheme. The heat conduction was calculated for a homogeneous snowpack of density 350 kg/m$^3$ with a grain size of 1 mm. Constant temperatures set to -15°C and 0°C were specified at the upper and lower boundaries, respectively. The snowpack was initially assumed to be isothermal at -10°C. Figure 1b shows the temperature distribution simulated by SMRFF against the results from OpenFOAM at three different times. The heat transport simulated by SMRFF proved to be in accordance with the CFD model, suggesting it will be adequate for heat flow simulations.





## 4 Flow finger simulation

In soil, PFPs were shown to develop in unsaturated porous media due to instabilities in the wetting front (Glass et al., 1989). The same process has been observed in snowpacks (e.g. Marsh, 1991) but little is known about the physical processes responsible for this instability. Marsh and Woo (1984a) observed that PFPs form in snow when the melt flow percolates to and
5 ponds at impeding layers, such as ice layers or at the interface of fine to coarse snow layers.

In the following sections, different processes that create wetting front instabilities in a snowpack will be demonstrated. For all the simulations in this section, the same snowpack with constant internal properties is used (Table 1).

### 4.1 Different numerical methods to simulate preferential flow paths

#### 4.1.1 Perturbation in mean snow grain size and density

Hirashima et al. (2014) demonstrated that PFPs can be simulated in a snow model when implementing a water entry pressure for dry snow, as well as spatial heterogeneities in snow grain size and density. However, their model was limited to an isothermal and homogeneous snowpack. Melt was not simulated, and only the infiltration of rain water was represented.

To trigger the formation of PFPs, a random perturbation ($\leq 1\%$) was applied at each numerical cell to the mean grain size, and then to the mean snow density chosen as inputs (Table 1). In Fig. 2, the water content distributions after 2 h of melt for
the two perturbed snowpacks are shown. Perturbing the snow density or grain size allowed the formation of an unstable wetting front, which then triggered PFPs.

The outflows from a snowpack (Table 1) with and without preferential flow paths (PFP) are shown in Fig. 3. In the case "without PFP" in Fig. 3, a horizontal flat wetting front propagated down the snowpack, and a pulse was observed when the wetting front reached the snow-soil interface. Then the outflow became constant over time as a steady state was reached. In
the case "with PFP" using a perturbation in snow density (Fig. 3), liquid water drained from the bottom of the snowpack earlier and pulses were observed in the computed outflow. These pulses were the result of the following pattern occurring within the snowpack after PFPs first formed: liquid water accumulated at the interface wet-dry snow layers, where PFPs were not present; when this liquid water reached a certain threshold, it flowed laterally due to lateral pressure gradients to where PFPs formed; finally, the water flowed down to the bottom of the snowpack through the PFPs.

#### 4.1.2 Perturbation in snow layer thicknesses

Another method to simulate the formation of PFPs consisted in implementing a random perturbation ($\leq 1\%$) to the snow layer thicknesses. The meltwater distribution after 2h of melt is shown in Fig. 2 under "Layer thickness". By implementing a random perturbation in the snow layer heights, the vertical distance between the centre points of two snow layers varied laterally, altering the magnitude of the vertical water fluxes (Eq. 4) between these two layers along the lateral length of the snowpack.
Liquid water initially flowed where the Darcy's flux first became positive, i.e. where the distances between two vertical neighbouring cells were smaller.



### 4.1.3 Spatial variability in heat flux and rain influx at the surface

In order to simulate an unstable wetting front, the heat flux or rain influx at the snow surface were laterally varied around their mean values ($\leq 1$ %). The water content inside the snowpack after 2h of simulation for a mean heat flux of 500 W/m$^2$ and a mean rain influx of 10 mm/hr is shown in Fig. 2 under "Heat flux" and "Rain flux", respectively. Doing so, the model was

able to simulate an unstable wetting front at the surface and therefore, PFPs.

### 4.2 Persistence and recurrence of preferential flow paths

Colbeck (1979), among others, observed that PFPs in snow persisted after forming. To show the ability of the model to simulate PFPs and their persistence, a simulation was run for 27h 45min with the following conditions: during the first third of the simulation, a rain influx of 10 mm/hr was applied at the surface (no melt occurred), then during the second third of the

simulation, no heat or rain flux were considered at the snow surface, allowing the liquid water that percolated the snow during the first period to drain out. Finally, during the last third of the simulation, a constant surface heat flux of 500 W/m$^2$ was used to melt the snow (no rain influx). Figure 4 shows the water content distribution within the snowpack at the middle of each phase. By the middle of the first phase, water from the rain flux percolated the snow and PFPs formed. Most of this liquid water had drained out of the snowpack at the middle of the second phase, and an equilibrium was almost attained where the

liquid water reached the irreducible water content. Finally, by the middle of the third phase, the "dry" PFPs from phase 2 were once again wet with meltwater from the snow surface. A horizontal wetting front was allowed to propagate at the upper 10 cm of the snowpack to better observe the persistence of the PFPs forming below it.

### 5 Sensitivity analysis

The sensitivity of the model inputs and parameters on its outflow after 2h of melt, the number of preferential flow paths

forming at the same time, the maximum liquid water content present in the snowpack  from the beginning of the melt and the time at which liquid water first reached the base of the snowpack were analysed.

### 5.1 Model inputs

A homogeneous snowpack was studied in this sensitivity analysis. Baseline properties characterising the reference snowpack are summarized in Table 1. For all the sensitivities, a perturbation of less than 1 % was applied independently to the mean

density and grain size. Model inputs were modified individually from the reference case to observe and compare their effects on model outputs. Figure 5 shows the model behaviours for different sensitivities of model inputs.

Snow density is the input that had the greatest effect on the maximum liquid water content in the snowpack. Indeed, higher snow density resulted in higher suction, which then caused more liquid water to accumulate at the interface wet to dry snow layers before Darcy's flux became positive so that water could flow downward. Due to this increase in liquid water, a higher

hydraulic conductivity was computed, resulting in faster mass flow and thus, increasing the outflow at the base of the



snowpack. On the contrary, the matric suction and water entry pressure for dry snow both decreased with larger grain (pore) size, reducing the liquid water content accumulating at the interface of wet to dry snow layers (Fig. 5). Because the hydraulic conductivity is greater for bigger grain (pore) sizes, the outflow increased and the time at which liquid water reached the base decreased with grain size.

The initial snow temperature and irreducible water content had great impacts on the timing of water reaching the base of the snowpack and therefore, on the snowpack outflow. A higher irreducible water content implies that more water is retained within the pores by capillary forces in the snow. This resulted in a delay in meltwater reaching the base of the snowpack and thus, in less outflow after 2h of melt. Similarly, a lower snowpack temperature caused more refreezing of liquid water in the snowpack, and thus delayed the timing of liquid water reaching the bottom of the snowpack.

**5.2 Model parameters**

A sensitivity analysis on the snow hydraulic parameters was conducted. A homogeneous snowpack with the properties presented in Table 1 was again considered for the reference case. Independent perturbations ($\leq 1\%$) in snow density and grain size were applied to simulate the formation of PFP. This analysis is important as all these parameters were determined from equations developed from laboratory experiments with a limited sample of snow properties. Measurement errors and data
fitting resulted in empirical equations with imperfect coefficients. This sensitivity analysis aimed to show how errors in estimating the empirical equations of the water entry pressure (Katsushima et al., 2013), the snow permeability (Calonne et al., 2012), for which the uncertainty of each coefficient was given by the authors (Eq. 5), and the two coefficients $\alpha$ and n in the van Genuchten model (Eq. 6 and 7) (Yamaguchi et al., 2012) impacted the model behaviour and outputs.

According to Fig. 6, the coefficients $\alpha$ and the first constant in the water entry pressure equation (Eq. 8) had the greatest
impact on the model. An increase in the first coefficient in the water entry pressure equation resulted in a higher water entry pressure in dry snow, which in turn caused liquid water to infiltrate for a lower water content accumulating at the interface wet to dry snow. This greatly impacted the number of PFPs forming in the snowpack. Less water within the snowpack resulted in lower hydraulic conductivities, thus in a lower outflow and a longer time for liquid water to reach the base of the snowpack. Shifts in the coefficient $\alpha$ had a similar impact on the model.

**6 Model application**

A model simulation of water and heat flows through a typical subfreezing, heterogeneous, layered snowpack is demonstrated as an example of model application. Random perturbations in grain size and density were introduced in each layer to simulate conditions that cause the formation of PFPs. The snowpack was divided into four horizontal snow layers (Table 2, Fig. 7). The third layer (from the bottom of the snowpack) is an ice layer of a higher density than the surrounding layers. Under natural
conditions, liquid water would accumulate over this layer and preferential flow paths form below this saturated horizontal layer (Marsh and Woo, 1984a).



Table 1 summarizes the parameters and inputs used in the model as initial and boundary conditions, and Table 2 presents the snow layering properties. The values used for the mean optical grain sizes in Table 2, which were assumed to be equal to the classical grain sizes for simplicity, were computed from the average specific surface areas measured by Montpetit et al. (2012) for different snow types.

5      The simulation was run until the snowpack completely melted. Figure 7 shows the water content, density, and temperature distributions within the snow layers at different times during melt. It can be observed that liquid water accumulated above the ice layer as well as above the lower layer, i.e. at the interface of fine to coarse snow layers. Then, preferential flows formed where smaller grain sizes were found in the impeding layers. Liquid water flowing through the preferential flow paths reached the base of the snowpack prior to the wetting front. Snow density slightly increased due to refreezing of liquid water, corresponding with an increase of snow temperature to the freezing point. While the snowpack was melting, zones within the snowpack stayed dry and cold until the main wetting front arrived or heat conduction increased the internal temperature. Heat conduction at the bottom of the snowpack, which temperature was set to 0°C, can be observed. As a result of heat conduction, cold snow areas around the PFPs (at freezing temperature) became warmer.

## 7 Discussion

15 As seen in Fig. 2, PFPs formed mainly at the surface of the snowpack as a result of instabilities in the wetting front. The water entry pressure for dry snow greatly impacted the accumulation of liquid water and the number of PFP forming. Indeed, a lower water entry pressure results in less accumulation of liquid water, and combined with heterogeneities in snow density and grain size, this results in less lateral heterogeneities in the wetting front. Therefore, the wetting front moves further downward before becoming unstable enough to create preferential flow.

20      Waldner et al. (2004) hypothesized that PFPs in snow can form due to heterogeneous infiltration patterns that can be caused by heterogeneities in the surface melt. From this hypothesis, spatial variations were introduced in the heat and rain fluxes at the snow surface. In nature, variations in the heat flux may be caused by terrain topography, vegetation, or cloudiness. Implementing perturbations in snow layer thicknesses for simulating PFPs is a numerical method to try to reproduce lateral heterogeneities in the properties of the horizon of snow layers as observed in the field by Wankiewicz (1979) and Marsh and 25 Woo (1984a).

The sensitivity analysis of the model inputs showed which snow properties should be carefully measured in the field to minimize errors in model prediction. It is shown that snow density, which is relatively easy to measure in the field, had a greater impact on model outflow than grain size which has great uncertainty in estimation. Marsh and Woo (1984b) showed with their calculations that an increase in snow density resulted in a decrease of the time at which liquid water reached the base 30 of the snowpack, and that the irreducible water content had an opposite effect. Their results are consistent with those shown in this paper, despite the different conditions used in both studies.


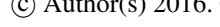


In the snow literature, many different values have been used for the irreducible water content in snow; for instance, Marsh and Woo (1984b) used a value of 0.07, whereas Hirashima et al. (2014) used 0.024; no relationship exists to relate this parameter to snow properties. This model shows that this parameter had a significant impact on the timing and quantity of snowmelt runoff. Runoff is also considerably impacted by the initial snow temperature before melt starts, as colder snowpacks

cause more refreezing and therefore delay the downward flux of water.

The purpose behind the sensitivity analysis on the empirical equations utilized to estimate the snow hydraulic parameters was to show the uncertainty behind the assumption that these equations are true for any snow type. Indeed, these empirical equations were developed from a limited number of artificial snow samples and the statistical regressions were biased. This model proved to be highly sensitive to two parameters, the variable $\alpha$ in the Van Genuchten model and the water entry pressure

via its first constant in Eq. 8.

The model application presented in Sect. 6 was able to reproduce processes observed in the field. Meltwater ponded above the ice layer as observed by Marsh and Woo (1984a) as well as at the interface between fine and coarse snow layers (Wakahama, 1974 and Waldner et al., 2004). Below these high saturation layers, preferential flow paths formed, which is in accordance with the field observations of Wankiewicz (1979) and Marsh and Woo (1984a). Moreover, PFPs acted as a way to

transfer energy from the snow surface to the snowpack internal energy. Indeed, temperature of the PFPs (at the freezing point) spread within the snowpack due to conduction, resulting in an increase of snow internal temperature. This is in accordance with the findings of Marsh and Woo (1984a). Under natural conditions, this temperature gradient would cause a vapour flux from the PFPs to the dry cold snow, causing grain growth (Colbeck, 1982) that would enlarge preferential flow paths.

## 8 Conclusion

The model developed here can simulate the formation and recurrence of PFPs that develop when a wetting front becomes unstable due to lateral heterogeneities in snow properties and layering.  It is based on recent snow physics studies and the results of soil physics studies over the last few decades and so estimates snow hydraulic and thermal properties with a high level of confidence. Different methods were applied to produce unstable wetting fronts, such as lateral heterogeneities in density, grain size, input fluxes at the surface or in snow layer thicknesses. The implementation of PFPs in a snow model

caused snowmelt runoff to occur sooner than considering a flat horizontal wetting front. Liquid water flowing through these preferential flow paths and arriving at the base of the snowpack prior to the wetting front generates basal ice layers when a subfreezing soil underlie the snow. The formation of PFPs proved to be important for the transfer of heat from the snow surface to the internal snow energy.  Moreover, the implementation of a water entry pressure for dry snow, used in this model due to a missing water retention curve for wetting snow, was a necessity for the formation of PFPs, as in Hirashima et al. (2014).

However, there are great uncertainties as of the empirical equation found to estimate this pressure. Indeed, this relationship depends solely on grain size and was developed using four artificial snow samples with densities greater than 387 kg/m$^3$.



Therefore, nothing is known on the applicability of this equation for snow types with lower densities and its dependence on snow density.

The sensitivity analysis on model inputs showed that density and snow temperature should be accurately measured in the field when used to initialize this model. Moreover, the irreducible water content, which value is inconsistent throughout the
snow literature, needs to be related to snow properties.

Lateral perturbations were used in the model to force the wetting front to become unstable; however, such perturbations cannot be implemented in one-dimensional snow models. As a result, 2D numerical snow models are a necessity to accurately predict the water flow through snow, resulting in better prediction of timing and quantity of snowpack runoff. Nonetheless, such a model could be parameterized, as done by Zhao and Gray (1997), to be implemented in hydrological platforms or land
surface schemes. In the model applications presented here, the spatial distribution of the perturbations was random. Thus, further work should be carried to establish relationships for the spatial distribution of these parameters from field observations.

In hydrological models or land surface schemes, it is assumed that a snowpack must be isothermal and wet for before runoff occurs. This assumption is erroneous as meltwater flows through PFPs, thus bypassing areas of the snowpack that stay dry. Also, a snowpack does not have to be isothermal for melt to start, only the upper layer, near the surface, must be isothermal at
0°C. Then, this is the meltwater penetrating deeper snowpack layers that will gradually warm the snowpack to the freezing point.

This two-dimensional snow model needs to be validated against *in-situ* data. This lack of validation makes this model only a tool to show how PFPs can be numerically reproduced, where the snow research is currently at, where the uncertainties are and what should be the next steps. Furthermore, this model has many assumptions that need to be addressed. Among these
assumptions, snow grain metamorphism is one of the biggest. Wet snow metamorphism would result in an increase of grain size where liquid water flows and dry snow metamorphism, caused by vapour gradients, could make the simulation of the preferential flow paths dynamic. The development of this numerical model raises questions on water flow through snow and numerical snow modelling:

- Does the irreducible water content depend on snow properties?
- How should the hydraulic conductivity and thermal conductivity be numerically computed at the interface of two numerical nodes?
- How should grain size and density perturbations be represented?
- How should the water entry pressure for dry snow be related to snow density?
- Is the thermodynamics equilibrium assumption suitable or can liquid water flows through subfreezing snow layers
without completely refreezing (Illangasekare et al., 1990)?
- Is the flow through preferential flow paths laminar? Does Darcy's law always apply?
- Can the equation used for the thermal conductivity in a dry snowpack (Eq. 13) be used when liquid water content is present within the snowpack?





- Does liquid water refreeze at 0°C or is there a freezing point depression that depends on snow properties and surface tension between ice and liquid water?
- Is convection between the wetting phase and the ice important for heat transfer within the snowpack?
- Is the dynamics of preferential flow paths caused by dry snow metamorphism generated by the temperature gradient around the "warm" PFP?

## Acknowledgements

Funding from the University of Saskatchewan, Natural Sciences and Engineering Research Council of Canada, Discovery and Research Tools and Instrumentation grants, Canada Foundation for Innovation, Canada Research Chairs, Global Institute for Water Security, NSERC Changing Cold Regions Network, and Alberta Agriculture and Forestry made this research possible.

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



**Table 1: inputs used for the simulations in Sect. 4, Sect. 5 and Sect. 6**

| | Section 4 | Section 5 | Section 6 |
|---|---|---|---|
| Snow type | Aged snow, faceted crystals | Aged snow, rounded crystals | cf. Table 2 |
| Horizontal length of snowpack | | 1 m | |
| Snow depth | | 1 m | |
| Number of horizontal cells | 50 | 45 | 50 |
| Number of vertical cells | 50 | 45 | 20 |
| Temperature at the interface snow-soil | | 0ºC | |
| Initial snow surface temperature | | 0ºC | -2 ºC |
| Initial snowpack temperature | | 0ºC | -2 ºC |
| Snowpack density | 350 kg/m$^3$ | 300 kg/m$^3$ | cf. Table 2 |
| Snowpack optical grain diameter | 1 mm | 0.3 mm | cf. Table 2 |
| Energy at the surface | | 500 W/m$^2$ | |
| Volumetric irreducible water content | | 0.02 | |
| Ground slope angle | | 0 | |



**Table 2: Snow layering system used in the application of Sect. 6**

|  | Type of snow | Thickness [m] | Temperature [ºC] | Density [kg/m³] | Optical grain diameter [mm] |
|---|---|---|---|---|---|
| Layer 1 (bottom) | Coarse depth hoar | 0.3 | -2 | 350 | 0.5 |
| Layer 2 | Dense rounded snow | 0.3 | -2 | 300 | 0.3 |
| Layer 3 | Ice layer | 0.1 | -2 | 450 | 0.7 |
| Layer 4 | Dense rounded snow | 0.3 | -2 | 300 | 0.3 |





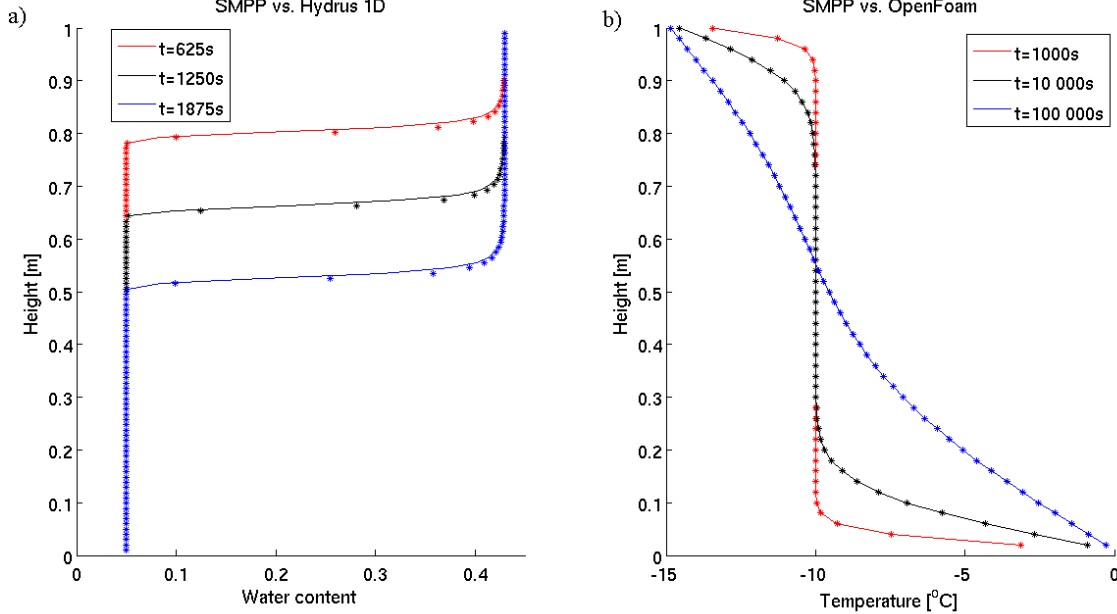

Figure 1: (a) Comparison of water content distributions at three different times simulated by SMPP (represented with the dots) and by the soil model Hydrus 1D (represented with the lines). (b) Comparison of temperature distributions at three different times estimated by SMPP (represented with the dots) and by the CFD model "OpenFOAM" (represented with the lines).





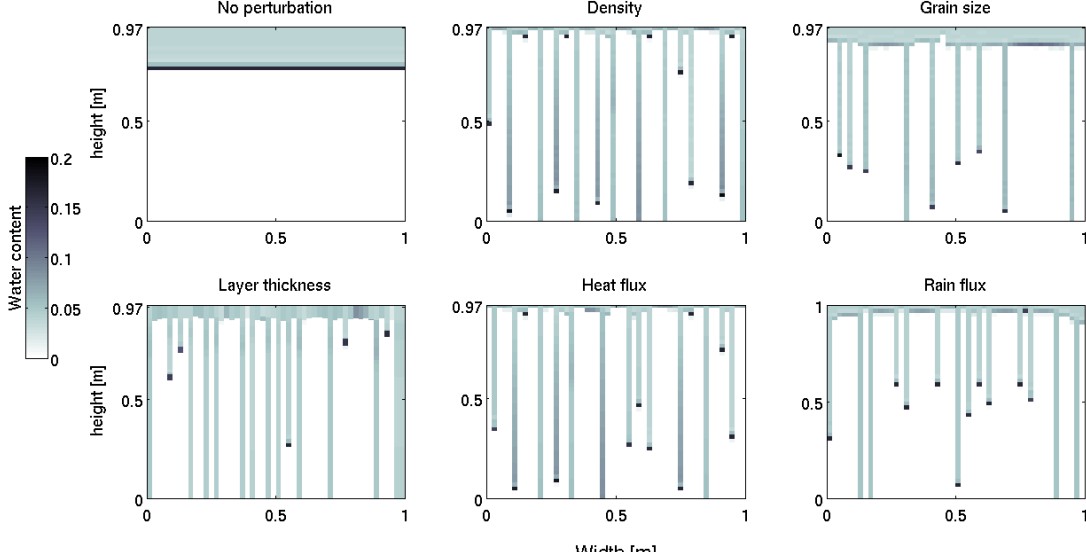

**Figure 2: Varying water distributions resultant of perturbations applied to different model inputs (titles of each figure) within a 1x1m single-layered isothermal snowpack (50x50 cells). The mean density and grain size are set to 350 kg/m³ and 1 mm, respectively.**



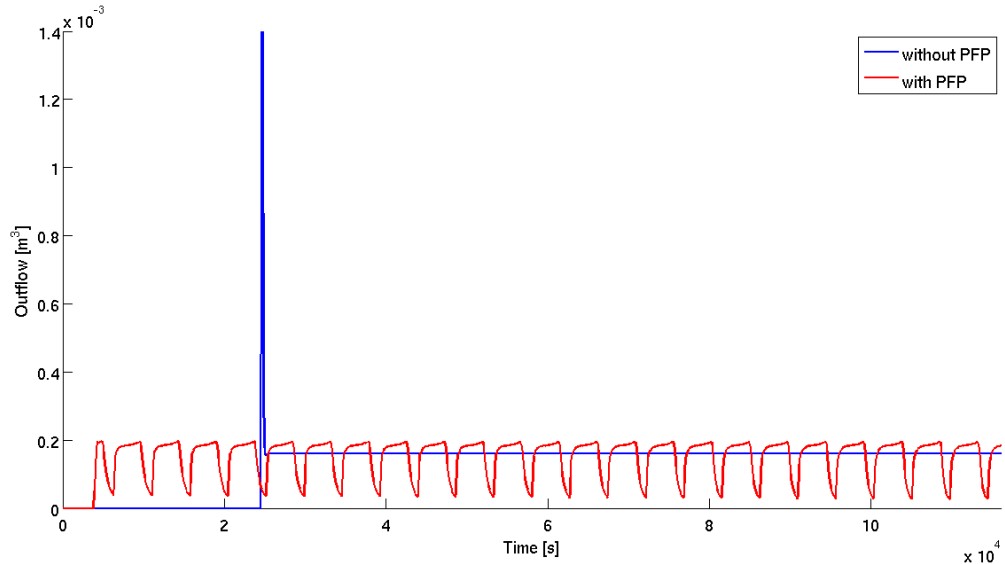

**Figure 3: Outflow computed by the model for two different cases: with and without perturbations ("with PFP" and "without PFP", respectively) applied to snow density to simulate the formation of preferential flow paths.**





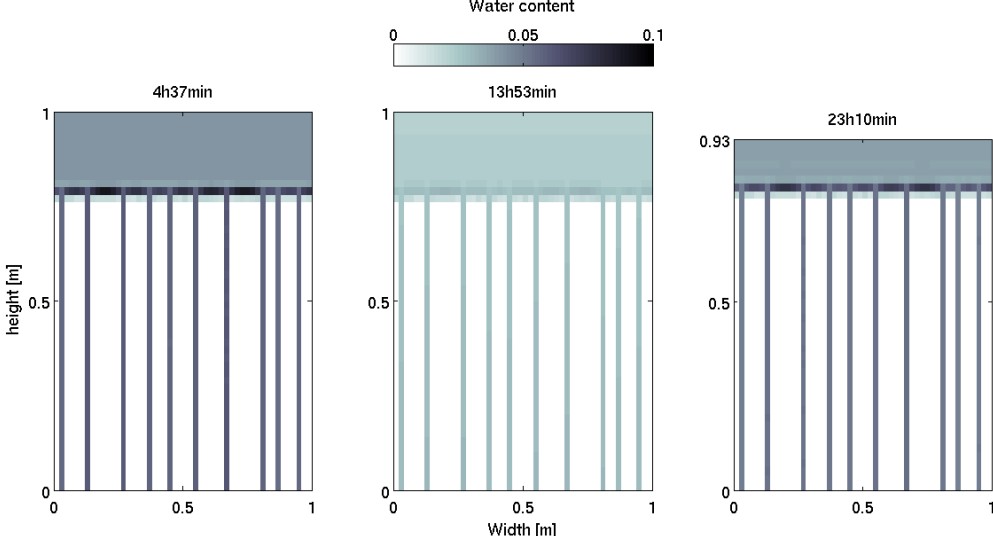

**Figure 4: Water content distribution at the middle of the first phase (4h37min), with rain influx at the surface, at the middle of the second phase (13h53min), with no flux at the snow surface and at the middle of the third phase (23h10min), with a heat flux applied at the snow surface.**





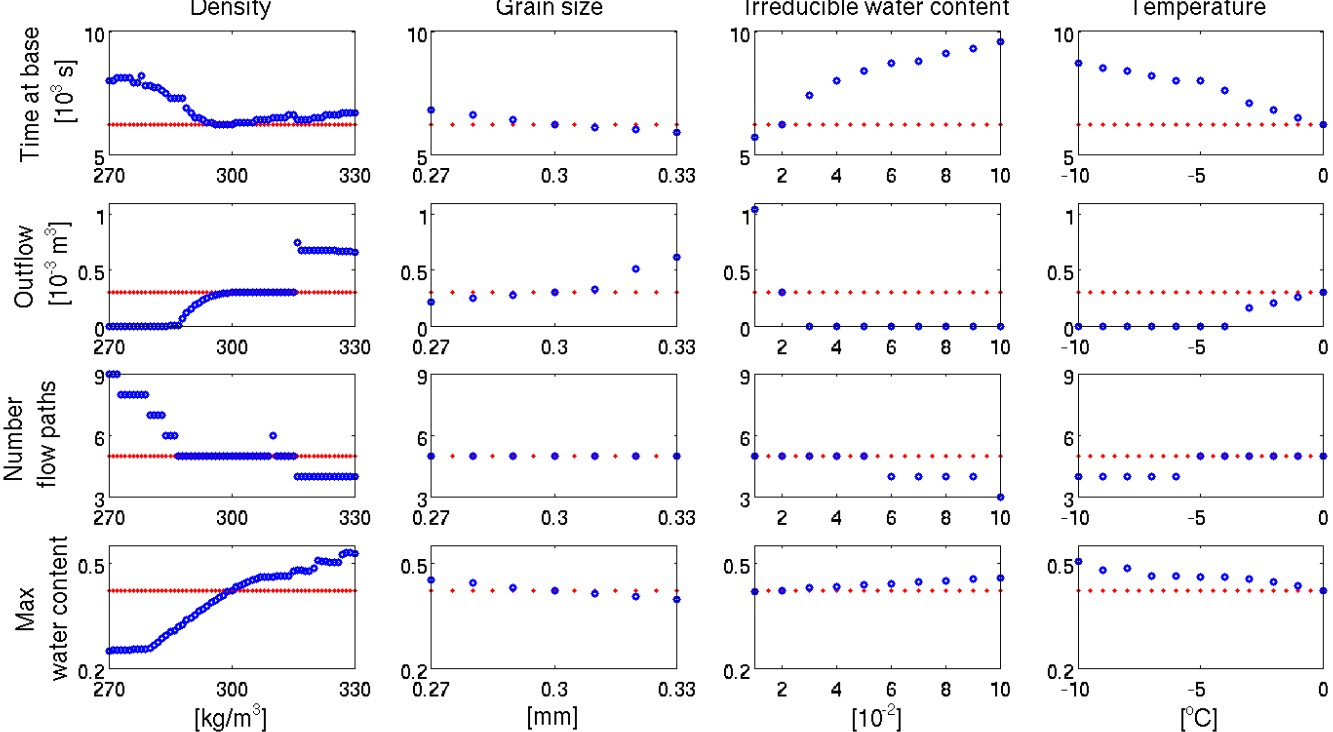

**Figure 5: Sensitivity analysis of model inputs (from left to right: mean dry density, grain diameter, irreducible water content, and snow temperature) on maximum water content in the snowpack ($m^3/m^3$) within the two first hours of melt (lower graphs), number of preferential flow paths forming ($\#/m^2$) after two hours of melt (second lower graphs), outflow from the snowpack after two hours of melt (third lower graphs) and time at which liquid water reaches the base of the snowpack (upper graphs). The red dots show the model outputs for the baseline reference case used for this sensitivity analysis (Table 1).**




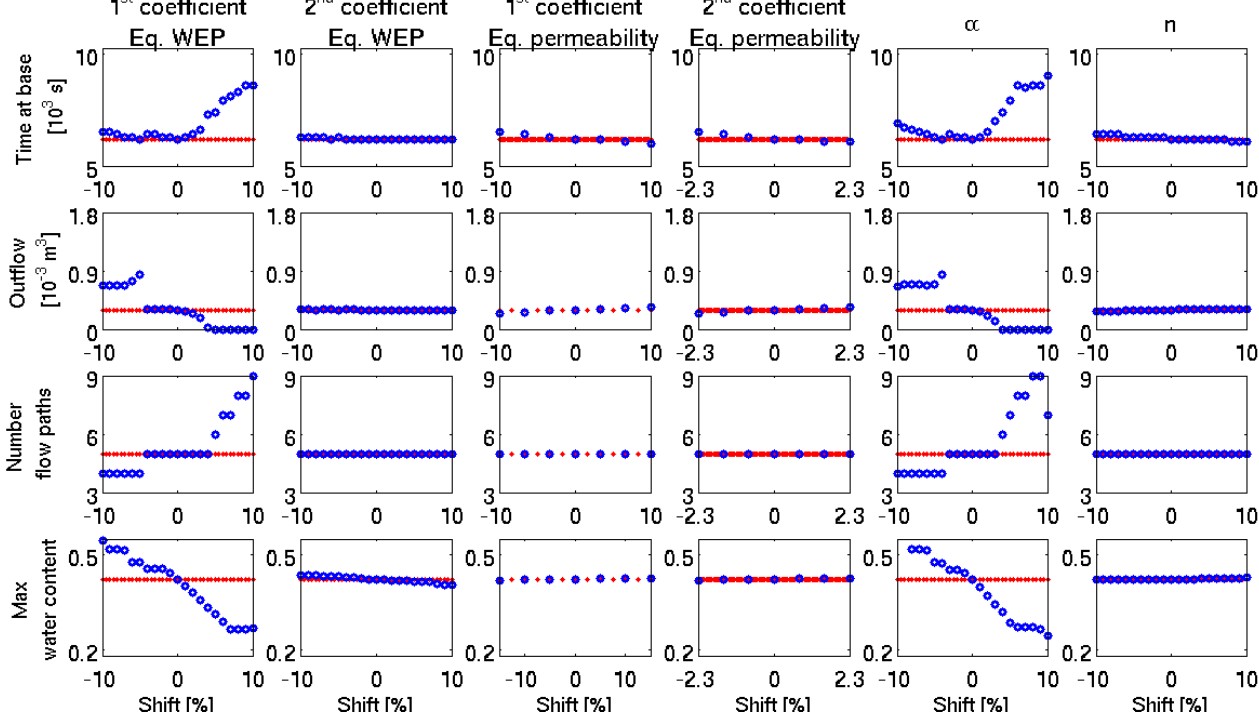

**Figure 6: Sensitivity analysis of coefficients in the empirical equations used to estimate snow hydraulic properties. From left to right, these coefficients are the two constants in the water entry pressure equation (Eq. 8), the two constants in the permeability equation (Eq. 5) and the coefficient s$\alpha$ and $n$ from the Van Genuchten model (Eq. 7). Analysis against the maximum volumetric water content (m³/m³) in the snowpack within the first two hours of melt (lower graphs), number of preferential flow paths (#/m²) forming after two hours of melt (second lower graphs), outflow from the snowpack after two hours of melt (third lower graphs) and time at which liquid water reaches the base of the snowpack (upper graphs). The red dots show the model outputs for the baseline reference case used for this sensitivity analysis (Table 1).**



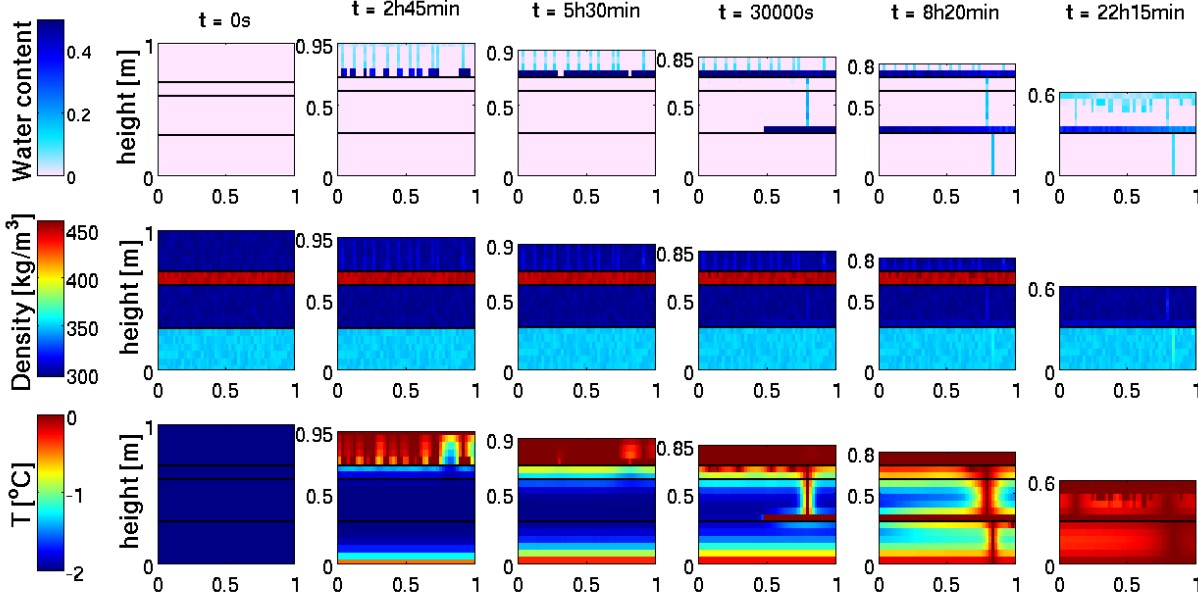

**Figure 7: Melt of a 1x1m layered snowpack. The upper graphs show the liquid water content distribution within the snowpack at different time steps, the middle graphs show the density within the snowpack at the same time steps and the lower graphs represent the temperature distribution within the initial cold snowpack. The horizontal dark lines represent the interface between each layer.**

