# Peer review of "A 2D model for simulating heterogeneous mass and energy fluxes through melting snowpacks"

_The Cryosphere, 2016_

## Referee Comment (RC1) · Anonymous Referee #1 · 14 Apr 2016

In this paper, authors developed new two-dimensional water transport model combining the processes of snow temperature change, snowmelt, refreezing and heterogeneous water transport model. Simulations of preferential flow considering the melt-freeze processes are very important, and this model has a potential to advance modeling studies of heterogeneous water infiltration in the cold snowpack. Components used in this model are basically theories in existence. Water infiltration schemes are almost same with Hirashima et al. (2014). Schemes of temperature and melt-freeze processes are already developed by Illungasekare (1990) and Daanen and Niever (2009). They also simulated interactions between liquid water and snow temperature.

Therefore, analysis of simulation results should show advantage of this combined

model and provide informative scientific results (e.g. enhanced accuracy or new simulation which cannot be performed by previous model). Authors showed many simulation results in sensitivity analysis, but discussions of sensitivity analysis were just confirm processes that were already known qualitatively. Furthermore, model application in section 6 did not apply to real snowpack observation data but only virtual snow stratigraphy. Due to lack of validation using real data, they could not show the accuracy of this model in the analysis quantitatively. Consequently, despite the model is innovative, this study could neither show availability to reproduce the real snowpack nor suggest additional experiment to improve the accuracy of the model sufficiently. Authors are not necessarily required to perform laboratory experiment or field observation by themselves, but in that case, they need to find any literature of real data to compare with the simulation results. If this model has new idea (e.g. new technic to compute quickly) or shows the new simulation that can be performed only by this model (for example, simulation of ice layer formation), this paper may make informative components even if simulation result is not compared with real data. In my opinion, although this model itself seems to be useful, authors should consider the direction of numerical analysis to produce informative scientific results.

minor comments

P3 L9 the model of Hirashima et al. (2014) is not limited to small artificial snow. Although their model neglected melt freeze processes, their model did not neglect multi layer. Simulations in that paper were performed in single layer snow because laboratory experiments were also performed using single layer column. They performed multi layer simulation in following proceedings although validation was not performed. Therefore, it should not be included as advantage in this model. You should replace with following sentence.

"However, their model was limited to isothermal snow samples, neglecting melting at the surface, and refreezing of liquid water."

1) Hirashima, H., S. Yamaguchi, and Y. Ishii, 2014. Simulation of liquid water infiltration into layered snowpacks using multi-dimensional water transport model. ISSW proceedings, 48-54.

2) Hirashima, H., S. Yamaguchi, and Y. Ishii, 2014, Application of a multi-dimensional water transport model to reproduce the temporal change of runoff amount. ISSW proceedings, 541-546.

P5 L10 Eq. (8) is not the equation of dE Rooji and Cho (1999). Katsushima et al (2013) found that the water entry suction of snow was about 1 cm larger than the estimated value by the equation of Baker and Hillel (2000) (hwe(m)=0.0437d^-1+0.00074). And then, Hirashima et al. (2014) added 0.01 in their equation. Furthermore, rc is half of d, so (1/2rc) is correct, not (2/rc).

P6 L25 How did you decided to use this boundary condition? What kind of situation were you going to reproduce? (e.g. For laboratory experiment, both right and left hand boundary should be no-flow boundary. For natural snow, both of them should be periodic boundary condition or free drainage boundary.)

P8L11-12 As mentioned in comment P3L9, Hirashima's model can consider multi snow layer. So it should be replaced with following sentence.

"However, their model was limited to an isothermal snowpack. "

P8 L17-25 Both runoff in the graph of Fig.3 are actually impossible. Graph without PFP is simulation result considering water entry suction without heterogeneity. This infiltration condition is different from matrix flow. In reality, the condition with completely homogeneous snow is impossible. Hirashima et al. (2014) showed the simulation of water infiltration in same condition in order to show that considering only water entry suction without heterogeneity is not sufficient to reproduce the preferential flow. The discussion of this impossible phenomenon does not have scientific signification. Graph with PFP also has problem. In the real condition with PFP, it is quite unlikely to occur
such a cyclic pulse in red graph. Isn't it just a fault of this model?

P9-10 Fig. 5 and 6. Irreducible water content, $\alpha$ and n value were determined from the water retention curve in laboratory experiment to optimize the curve. Thus, these values are linked to other parameters each other. Therefore, individual sensitivity experiment with static values of two parameters does not have scientific signification to describe the effect of estimation error. Sensitivity analysis for snow temperature has a potential to show the scientific informative result using this model. However, this result just showed that the low snow temperature leads to delay of runoff by refreezing. It lacks the impact to show advantage of this model.

P10-11 Fig. 7 Applying new numerical model for natural snow is beneficial. However, it was not applied to real data obtained by snowpack observation but to virtual snow layer. If this model applied to real snow using snowpack observations and simulate water infiltration for the duration of interval of two snowpack observations, simulation result can be compared with the real data. If this model can, I expect the reproduction simulation of ice layer formation in the snowpack in this model.

Overall, although the developed model itself is advanced numerical water transport model, numerical analysis could neither show the advantage nor accuracy of this model. Discussion without any validation by real data could lead erroneous opinion such as the case of runoff in Fig. 3. Furthermore, it is necessary to perform numerical analysis to provide scientific informative result. I believe that if this model is validated using real data and show reproduction simulation of ice layer formation, this model can provide scientific informative results.

---

## Referee Comment (RC2) · Anonymous Referee #2 · 28 Apr 2016

The paper describes a 2D snowpack model, that solves heat and mass equations (Richards equation), in order to assess the liquid water flow in snow, with a particular focus on preferential flow. The model can be regarded as an experimental model, as some important processes found in natural snow covers are not represented (i.e., snow settling or snow metamorphism), which is not an immediate problem for the study presented in the paper. The novelty in the model approach is a coupling between heat and mass equations (taking into account phase changes), whereas the principles behind, and description of, the formation of preferential flow in snow are mostly known from earlier studies. Regarding this point, it basically is a re-implementation of previous studies. This is not necessarily a problem, as independent verification of results

is a central part of science. However, my feeling is that the paper in some aspects just falls short of providing the important results that could be achieved by using the model described by the authors. Two things come in mind: first, some comparison with field data or laboratory data (there is a lot out there in datasets or publications of laboratory experiments) to validate the model results with "real" snow covers. A second alternative route is to better set up the sensitivity study, such that a connection with natural snow covers is made. I will provide some extra explanation regarding this point below. But to summarize, it is basically not clear whether the sensitivity study was supposed to span the typical variations of snow cover properties in natural snow covers, or is representing typical measurement errors. So in its current form, the manuscript is neither providing the validation with field data, nor those results from the sensitivity experiments that may serve other researchers. Furthermore, I do like concise papers, and this paper is generally concisely and pleasantly written, but in many crucial details, its lacking the necessary information to understand the work (see my comments below). Nevertheless, the study contains relevant results, and is also timely (the interest in liquid water flow in snow seems to have a new boost since a few years), potentially fitting well in ongoing studies and discussions. However, in my opinion, it requires a thorough revision in order to make the manuscript more mature for publication.

I actually think that the authors did do a very good job in constructing the numerical model, which was probably not easy to achieve! I can imagine that it involved some hard and dedicated work, and I think that the model is forming a solid framework for future studies. In this light, I would like to encourage the authors to release the source code as open source to the community. This would also further enhance the importance of the study.

Major comments:

- The authors apply random perturbations to the snow cover properties in the order of $1\%$. There is no motivation provided by the authors why this value was chosen. Also no information about the procedure to apply the perturbations is provided (is it

Gaussian?). It seems a significantly smaller perturbation than used by Hirashima et al. (2014) and, more importantly, Hirashima et al. (2014) found that the results are strongly dependent on the applied perturbation! Right now, this is not at all discussed in the manuscript.

- I have doubts whether Eq. 8 is correct. Actually, it seems to be very similar to Eq. 15 in Hirashima et al. (2014), where it is a modification of the equation found by (Baker and Hillel, 1990), see Katsushima et al. (2013) for details. So I wonder whether the provided citation (Rooji and Cho (1999)) by the authors is correct. Furthermore, given that diameter = 2*radius, the 2 in Eq. 8 should be in the denominator, following the equation provided by Hirashima et al. (2014).

- A similar problem is found in Eq. 7, where I think the factor 2 should be in the denominator. It is recommended that the authors verify their implementation in the code, as the simulations may change considerably for these type of errors.

- Section 3.2: please make a distinction between approximations and unknowns. For example: I think it is not justified to claim that changes in grain size were not simulated, because (L3) "due to the lack of complete understanding of the physics of these processes, ... The assumptions made in this model also indicate the current knowledge...", because the SNOWPACK model for example is simulating grain growth in the presence of water based on the results by Brun et al. 1989b (INVESTIGATION ON WET - SNOW METAMORPHISM IN RESPECT OF LIQUID- WATER CONTENT), Ann. of Glaciol. 13. So I do understand that in this version of the model, the authors neglect grain growth, but in my opinion, it is a misrepresentation to claim that it is necessary due to the lack of understanding. Similar for point 3. This assumption is made for convenience. I can agree with the assumption, but it should not be implied that this is due to the lack of understanding.

- Section 4.2 is for me a bit problematic. Colbeck (1979) indeed found that preferential flow paths in snow persisted after forming. This probably is due to changes in

snow microstructure. A counter example is provided by Schneebeli (1995), a reference which deserves citation here. Using dye tracer, he found that actually preferential flow paths are not (necessarily) constant in space and time. Right now, section 4.2 in the manuscript is actually missing a kind of concluding remark, but it sounds like the authors claim that their model reproduces persistence in the PFP. But in my opinion, the persistence of the preferential flow paths in their model is likely there, because the random perturbations did not change, as the model by the authors do not have a microstructure model in their snow model. Then it is a kind of: "getting the right results for the wrong reason". So I think the section may need to be removed from the manuscript, or else it should be much better defended why the model by the authors is congruent with the observations by Colbeck (1979), and not those by Schneebeli (1995). Which part of the physics in the model is confirming the result by Colbeck (1979) and not the result by Schneebeli (1995)? Reference: MARTIN SCHNEEBELI Biogeochemistry of Seasonally Snow-Covered Catchments (Proceedings of a Boulder Symposium July 1995). IAHSPubl.no. 228,1995. Development and stability of preferential flow paths in a layered snowpack.

Major comments regarding sensitivity study:

- It is not well motivated where the sensitivity study is based on. For example, snow density is varied by 10%, which one can regard as the typical accuracy with which snow density can be determined in the field. However, the range of densities found in a natural snow cover range from roughly 100 kg/m$^3$ for new snow to 400-500 kg/m$^3$ for old snow and up to 600 kg/m$^3$ for firn. So here, the sensitivity study seems to capture measurement error rather than the range of values found in natural snow covers. On the contrary, the sensitivity study for temperature ranges over 10 degrees. This is the opposite, rather capturing the natural variability found in snow covers than measurement errors.

- A similar comment can be made about the sensitivity study for $\alpha$ and $n$. Where is the choice of a variation of +/- 10% based on? As $\alpha$ and $n$ are coupled via $\rho/d$ (see

Yamaguchi et al. 2012), it is doubtful whether it is an informative result to vary both coefficients separately. I think in the end it is important how much the water retention curve changes. When n is small (1-2), 10% causes a big change in water retention curve. When n is $> 5$, the effect is much smaller. Opposite with $\alpha$. When $\alpha$ is large, a 10% has more influence then when $\alpha$ is small. So just modifying $\alpha$ and $n$ independently, for just one value of $\rho/d$, is not so informative.

- Also grain size is varied over only a very small range. However, grain size has a very important effect on the area that is involved in preferential flow (see for example Katsushima et al. (2013) and Hirashima et al. (2014)). Can the SMPP model reproduce these results?

- I can understand the confusion with the irreducible water content. It is true that a similar term is used in the Marsh and Woo papers (1984a,b), although they use the term saturation. It is also true that they used a value of 0.07. However, I do not think that this value is comparable to the role of the residual water content in the water retention curve, where it basically is the lower asymptote of dry conditions. My interpretation of the value used by Marsh and Woo is that the irreducible water saturation is actually the value of $S_w$ (the water saturation) in Equation 1 in the Marsh and Woo (1984a) paper. The saturation is defined between 0 and 1 where 0 is dry snow (or to be precise, snow at residual water content), and 1 is all pores are filled. That means that is should be scaled with the porosity to get the volumetric water content, which would be comparable to the residual water content as used by Yamaguchi et al. (2012). Assume a typical porosity of 0.6 for snow, the irreducible water saturation would translate into a irreducible water content (volumetric) of around 4%. Furthermore, my interpretation of their definition of irreducible is more in a bucket type approach, i.e., a typical amount of liquid water that remains in the pores without significant amounts of water flowing, which is not necessarily equal to the dry limit of the van Genuchten water retention curve. In a bucket scheme, typically a value of 4% is used (see for example Wever et al. 2014). Note that field measurements of bulk liquid water content typically

ranges from 0.02-0.04 (see Heilig et al. (2015), a reference that deserves citation in this manuscript).

Although I also think that the residual water content as used in the water retention curve is likely grain shape and/or grain size dependent, the range used in the sensitivity study (1-10%) doesn't seems to be realistic, given the observational evidence in literature. This is again an example where the choice of range for the sensitivity study is not well motivated, and is actually much larger than for snow density, given the typical range for these properties you will find in nature.

As the reference list provided by the authors is not clear at this point, the 1984a paper is for me: Wetting Front Advance and Freezing of Meltwater Within a Snow Cover 1. Observations in the Canadian. Reference: Heilig, A., C. Mitterer, L. Schmid, N. Wever, J. Schweizer, H.-P. Marshall, and O. Eisen (2015), Seasonal and diurnal cycles of liquid water in snow - Measurements and modeling, J. Geophys. Res. Earth Surf., 120 (10), 2139-2154, doi: 10.1002/2015JF003593.

Minor comments:

- Figure 3 is not really informative, as it is not at all clear if the change in runoff with or without preferential flow has any correspondence with reality. Moreover, the choice to only show the result for density is somewhat arbitrary.

- Figure 7: the 4th column of graphs, the title is suddenly expressing the time in seconds, not in hours/minutes.

- P3L17: "A melting snow cover can be considered a moving boundary"

- P2L28. Should this not read "the minimum suction"? I guess it depends on how positive/negative suction is defined?

- Eq 3 is not Richards equation, but just mass conservation. Richards equation is combining mass conservation with Darcy-Buckingham's law. Eq. 3 is valid under many more definitions of the flux q, of which Richards equation is a special category.

- Note that Darcy's law is basically the formulation for saturated flow, where the Darcy-Buckingham's law is valid for variable saturation, by introducing a water contents dependence on the hydraulic conductivity.

- P4: to be precise: Calonne et al. (2012) developed a relationship for permeability, which can be translated in saturated hydraulic conductivity.

- P4, L20: "is" -> "describes"

- P4: Eq 6 should be placed after L20-21.

- P4L25-26 should move to another place, as first the Equations need to be introduced.

- P5L13-14: It seems that here, dry snow is defined when volumetric LWC is below residual water content, and wet when it is above. Yamaguchi's formulation (i.e., the van Genuchten water retention curve) as far as I know is not applicable at all when LWC is below residual water content. So I don't understand this sentence.

- P6L15: "optimum grid size". Please provide the value for optimum grid size here. I'm also confused why there is no mention about the time steps? Convergence is often determined by both the time step and the grid cell size. Maybe also mention on what type of computer the simulations are run, and how much CPU time is needed for certain simulations, to give the reader an idea of the computational requirements of the model.

- Section 3.1 is confusing. It sounds as if the snowpack could be considered as being on a slope, where the left-hand side is upslope, and the right hand side is downslope (thus the specific choice of boundary conditions), but this is not explicitly explained and is a bit a puzzle right now.

- P6L26: Actually, free drainage boundary conditions are a type of Neumann boundary conditions.

- P6L27: I'm confused about the slope angle. When I'm correct, no result is shown that depends on slope angle? All the results seem to be for a flat snowpack. Furthermore,

Eq. 4 is only valid for flat conditions, or when the snowpack is considered vertically (which makes the description of the boundary conditions a bit more complicated). Often in snowpack simulations, the snowpack is considered slope-perpendicular, in which case Eq. 4 needs a modification for the slope angle. Of course, it all depends on definitions of for example the z-coordinate. In any case, the manuscript is confusing at this point and some more clarification is needed.

- The issue which is addressed on P13L25 seems to be linked to the numerical scheme, but is also not addressed in the appropriate section. So now this point comes out of the blue in the conclusions.

References:

- Please provide DOIs, should be standard nowadays!

- de Rooji: should read de Rooij.

- The difference between Marsh and Woo, 1984a and b is not made in the reference list. Which one is which?

- The paper describing SNOWPACK, part II, is having the wrong author list.

- At least one reference is missing, which is cited in the text (Wever et al. 2014).

---

## Author Comment (AC1) · 13 Jun 2016

Reply to anonymous Referee #1

The authors thank Referee #1 for the detailed comments below and the decision for minor comments.

In this paper, authors developed new two-dimensional water transport model combining the processes of snow temperature change, snowmelt, refreezing and heterogeneous water transport model. Simulations of preferential flow considering the melt-freeze processes are very important, and this model has a potential to advance modeling studies of heterogeneous water infiltration in the cold snowpack. Components used in this model are basically theories in existence. Water infiltration schemes are almost same with Hirashima et al. (2014). Schemes of temperature and melt-freeze processes are already developed by Illangasekare (1990) and Daanen and Niever (2009). They also simulated interactions between liquid water and snow temperature.

Therefore, analysis of simulation results should show advantage of this combined model and provide informative scientific results (e.g. enhanced accuracy or new simulation which cannot be performed by previous model).

- The components of this model are based on verified theories, but they have never been compiled comprehensively before. Therefore this is a new model with a unique range of capabilities and formulations that presents the importance of coupling heat transfer and water flow through both snow matrix and preferential flow paths. Through the application presented in Section 6, it was shown that preferential flow paths have a significant impact on the warming phase of the snowpack. Section 6 has been enhanced to show the potential of the model to simulate ice layers by varying the heat flux at the surface through time to represent a melt-freeze cycle. This capability is unique amongst snow models and simulates an important natural phenomenon for the first time.

Authors showed many simulation results in sensitivity analysis, but discussions of sensitivity analysis were just confirm processes that were already known qualitatively.

- The sensitivity analysis has been modified and is now discussed more deeply.

Furthermore, model application in section 6 did not apply to real snowpack observation data but only virtual snow stratigraphy. Due to lack of validation using real data, they could not show the accuracy of this model in the analysis quantitatively. Consequently, despite the model is innovative, this study could neither show availability to reproduce the real snowpack nor suggest additional experiment to improve the accuracy of the model sufficiently. Authors are not necessarily required to perform laboratory experiment or field observation by themselves, but in that case, they need to find any literature of real data to compare with the simulation results.

- Aspects of this model's matrix flow have been verified against the detailed model Hydrus 1D and the heat flow equation has been verified against a solver from OpenFoam. However, the authors agree that the model lacks of validation against observed data. Therefore, this paper only presents qualitative results and demonstration of outputs that are qualitatively similar to observations such as dye tracing cross-sections and snowpit wetness and temperature profiles. A full validation of the model will be performed in a future study.

If this model has new idea (e.g. new technic to compute quickly) or shows the new simulation that can be performed only by this model (for example, simulation of ice layer formation), this paper may make informative components even if simulation result is not compared with real data. In my opinion, although this model itself seems to be useful, authors should consider the direction of numerical

analysis to produce informative scientific results.

- This model has proven to be robust and stable despite the complexity of numerically coupling the heat and mass fluxes. Moreover, the model is proven to conserve mass thanks to the use of the finite volume method to discretize the partial differential equations. A greater discussion of these features is now added to the manuscript.
- The ability of the model to reproduce ice layer formation has been added to the manuscript and demonstrated.

Minor comments from Referee #1:

P3 L9 the model of Hirashima et al. (2014) is not limited to small artificial snow. Although their model neglected melt freeze processes, their model did not neglect multilayer. Simulations in that paper were performed in single layer snow because laboratory experiments were also performed using single layer column. They performed multi layer simulation in following proceedings although validation was not performed. Therefore, it should not be included as advantage in this model. You should replace with following sentence.
"However, their model was limited to isothermal snow samples, neglecting melting at the surface, and refreezing of liquid water."

1) Hirashima, H., S. Yamaguchi, and Y. Ishii, 2014. Simulation of liquid water infiltration into layered snowpacks using multi-dimensional water transport model. ISSW proceedings, 48-54.
2) Hirashima, H., S. Yamaguchi, and Y. Ishii, 2014, Application of a multi-dimensional water transport model to reproduce the temporal change of runoff amount. ISSW proceedings, 541-546.

- The sentence has been revised and the above conference proceeding citations included in the revised manuscript.

P5 L10 Eq. (8) is not the equation of de Rooji and Cho (1999). Katsushima et al (2013) found that the water entry suction of snow was about 1 cm larger than the estimated value by the equation of Baker and Hillel (2000) (hwe(m)=0.0437d^-1+0.00074). And then, Hirashima et al. (2014) added 0.01 in their equation. Furthermore, rc is half of d, so (1/2rc) is correct, not (2/rc).

- The citation "de Rooij and Cho (1999)" was not used for Eq. 8 but for the equation of air entry pressure in the model (not presented in the manuscript). The sentence P5 L10 has been moved to the end of Section 2.2 to avoid further confusion.
- Equations 7 and 8 have been corrected.

P6 L25 How did you decided to use this boundary condition? What kind of situation were you going to reproduce? (e.g. For laboratory experiment, both right and left hand boundary should be no-flow boundary. For natural snow, both of them should be periodic boundary condition or free drainage boundary.)

- Different boundary conditions can be chosen for the left and right hand boundaries in this model: periodic, both no-flow, and no-flow and free-flow for the left and right boundaries, respectively.  The authors chose the third option as an example but we agree that it is more appropriate to use no-flow boundary conditions for the two lateral boundaries in this level case. Section 3.1 has been modified to specify that the lateral boundaries are no-flow boundary conditions for this model application.

P8 L11-12 As mentioned in comment P3L9, Hirashima's model can consider multi snow layer. So it should be replaced with following sentence. "However, their model was limited to an isothermal

snowpack. "

- Corrected

P8 L17-25 Both runoff in the graph of Fig.3 are actually impossible. Graph without PFP is simulation result considering water entry suction without heterogeneity. This infiltration condition is different from matrix flow. In reality, the condition with completely homogeneous snow is impossible. Hirashima et al. (2014) showed the simulation of water infiltration in same condition in order to show that considering only water entry suction without heterogeneity is not sufficient to reproduce the preferential flow. The discussion of this impossible phenomenon does not have scientific signification. Graph with PFP also has problem. In the real condition with PFP, it is quite unlikely to occur such a cyclic pulse in red graph. Isn't it just a fault of this model?

- The purpose of this graph was to show the difference in model outflow between considering the formation of PFP or not. It was not to represent natural conditions.
- The cyclic pulse was not a fault of this model but is explained by the fact that matrix flow feeds PFP through lateral flows, as explained in Jury et al. (2003).
- As it seems that Fig.3 can be misinterpreted (c.f. comments from Referee #2), it has been deleted in the revised manuscript.

Reference:
Jury, W., Wang, Z., and Tuli, A. : A conceptual model of unstable flow in unsaturated soil during redistribution, Vadose Zone Journal, 2, 61–67, 2003.

P9-10 Fig. 5 and 6. Irreducible water content, $\alpha$ and $n$ value were determined from the water retention curve in laboratory experiment to optimize the curve. Thus, these values are linked to other parameters each other. Therefore, individual sensitivity experiment with static values of two parameters does not have scientific signification to describe the effect of estimation error. Sensitivity analysis for snow temperature has a potential to show the scientific informative result using this model. However, this result just showed that the low snow temperature leads to delay of runoff by refreezing. It lacks the impact to show advantage of this model.

- The authors agree. The range used for the sensitivity analysis on the irreducible water content has been changed to [0.01,0.04] (range observed by Katsushima et al., 2013). As for the sensitivity analysis on the parameters $\alpha$ and $n$, it has been replaced by a sensitivity analysis on the three different algorithms available for these parameters: Daanen and Nieber (2009), Yamaguchi et al. (2010) and Yamaguchi et al. (2012).
- The sensitivity analysis on snow initial temperature has now been more deeply discussed.

P10-11 Fig. 7 Applying new numerical model for natural snow is beneficial. However, it was not applied to real data obtained by snowpack observation but to virtual snow layer. If this model applied to real snow using snowpack observations and simulate water infiltration for the duration of interval of two snowpack observations, simulation result can be compared with the real data. If this model can, I expect the reproduction simulation of ice layer formation in the snowpack in this model.

- The formation of ice layer with this model is now shown and discussed.

Overall, although the developed model itself is advanced numerical water transport model, numerical analysis could neither show the advantage nor accuracy of this model. Discussion without any validation by real data could lead erroneous opinion such as the case of runoff in Fig. 3. Furthermore, it is necessary to perform numerical analysis to provide scientific informative result. I believe that if this model is validated using real data and show reproduction simulation of ice layer formation, this model can provide scientific informative results.

- The meaning of "numerical analysis" and its implication for this paper is not clear. The authors now specify on what CPU the model is run and the time it takes to run the simulations. The authors also further discuss the stability and mass conservation features of the model.

---

## Author Comment (AC2) · 13 Jun 2016

Reply to Referee no 2:

The authors thank Referee #2 for the detailed comments below.

The paper describes a 2D snowpack model that solves heat and mass equations (Richards equation), in order to assess the liquid water flow in snow, with a particular focus on preferential flow. The model can be regarded as an experimental model, as some important processes found in natural snow covers are not represented (i.e., snow settling or snow metamorphism), which is not an immediate problem for the study presented in the paper. The novelty in the model approach is a coupling between heat and mass equations (taking into account phase changes), whereas the principles behind, and description of, the formation of preferential flow in snow are mostly known from earlier studies. Regarding this point, it basically is a re-implementation of previous studies. This is not necessarily a problem, as independent verification of result is a central part of science. However, my feeling is that the paper in some aspects just falls short of providing the important results that could be achieved by using the model described by the authors. Two things come in mind: first, some comparison with field data or laboratory data (there is a lot out there in datasets or publications of laboratory experiments) to validate the model results with "real" snow covers.

- The validation of the model against field data will be done in a future study. To achieve such work, the energy balance at the snow surface needs to be implemented in the model, which is not the case in this version. It will be added in the next version of the model.

A second alternative route is to better set up the sensitivity study, such that a connection with natural snow covers is made. I will provide some extra explanation regarding this point below. But to summarize, it is basically not clear whether the sensitivity study was supposed to span the typical variations of snow cover properties in natural snow covers, or is representing typical measurement errors.

- The sensitivity analysis has been modified to account for the typical variations of snow cover properties in natural snow covers. See comments below.

So in its current form, the manuscript is neither providing the validation with field data, nor those results from the sensitivity experiments that may serve other researchers. Furthermore, I do like concise papers, and this paper is generally concisely and pleasantly written, but in many crucial details, its lacking the necessary information to understand the work (see my comments below). Nevertheless, the study contains relevant results, and is also timely (the interest in liquid water flow in snow seems to have a new boost since a few years), potentially fitting well in ongoing studies and discussions. However, in my opinion, it requires a thorough revision in order to make the manuscript more mature for publication.

I actually think that the authors did do a very good job in constructing the numerical model, which was

probably not easy to achieve! I can imagine that it involved some hard and dedicated work, and I think that the model is forming a solid framework for future studies. In this light, I would like to encourage the authors to release the source code as open source to the community. This would also further enhance the importance of the study.

- We thank the referee for this positive comment.
- After validation of model against field data in a near future study, the authors will release the source code as open source.

Major comments:

- The authors apply random perturbations to the snow cover properties in the order of 1%. There is no motivation provided by the authors why this value was chosen. Also no information about the procedure to apply the perturbations is provided (is it Gaussian?). It seems a significantly smaller perturbation than used by Hirashima et al. (2014) and, more importantly, Hirashima et al. (2014) found that the results are strongly dependent on the applied perturbation! Right now, this is not at all discussed in the manuscript.

- The perturbation used in this manuscript is smaller than in Hirashima et al. (2014) as the scales are different. In Hirashima et al. (2014), each numerical cell had a length of 5mm. In our model, they are at least an order of magnitude larger. The perturbations in snow cover properties is assumed to become smaller as the scale increases, much as slope declines in DEMs as resolution decreases. The selection of 1% was guided by the concept that it is reduced with model resolution. The perturbation is Gaussian and this has now benn described in the manuscript more completely.

- I have doubts whether Eq. 8 is correct. Actually, it seems to be very similar to Eq. 15 in Hirashima et al. (2014), where it is a modification of the equation found by (Baker and Hillel, 1990), see Katsushima et al. (2013) for details. So I wonder whether the provided citation (Rooji and Cho (1999)) by the authors is correct. Furthermore, given that diameter = 2*radius, the 2 in Eq. 8 should be in the denominator, following the equation provided by Hirashima et al. (2014).

- The citation de Rooji and Cho (1999) was not used for the water entry pressure equation but for the air entry pressure equation that is used in the model. Seeing as this citation was confusing, this sentence has been moved to the end of the section.

- Eq. (8) has been corrected. In the code, it is correct.

- A similar problem is found in Eq. 7, where I think the factor 2 should be in the denominator. It is recommended that the authors verify their implementation in the code, as the simulations may change considerably for these type of errors.

- Eq. 7 has been corrected. The code was correct as grain diameter is used.

- Section 3.2: please make a distinction between approximations and unknowns. For example: I think it is not justified to claim that changes in grain size were not simulated, because (L3) "due to the lack of complete understanding of the physics of these processes, ... The assumptions made in this model also indicate the current knowledge...", because the SNOWPACK model for example is simulating grain growth in the presence of water based on the results by Brun et al. 1989b (INVESTIGATION ON WET – SNOW METAMORPHISM IN RESPECT OF LIQUID- WATER CONTENT), Ann. of Glaciol. 13. So I do understand that in this version of the model, the authors neglect grain growth, but in my opinion, it is a misrepresentation to claim that it is necessary due to the lack of understanding. Similar for point 3. This assumption is made for convenience. I can agree with the assumption, but it should not be implied that this is due to the lack of understanding.

- This section has been clarified to distinguish between approximations and unknowns.

- Section 4.2 is for me a bit problematic. Colbeck (1979) indeed found that preferential flow paths in snow persisted after forming. This probably is due to changes in snow microstructure. A counter example is provided by Schneebeli (1995), a reference which deserves citation here. Using dye tracer, he found that actually preferential flow paths are not (necessarily) constant in space and time. Right now, section 4.2 in the manuscript is actually missing a kind of concluding remark, but it sounds like the authors claim that their model reproduces persistence in the PFP. But in my opinion, the persistence of the preferential flow paths in their model is likely there, because the random perturbations did not change, as the model by the authors do not have a microstructure model in their snow model. Then it is a kind of: "getting the right results for the wrong reason". So I think the section may need to be removed from the manuscript, or else it should be much better defended why the model by the authors is congruent with the observations by Colbeck (1979), and not those by Schneebeli (1995). Which part of the physics in the model is confirming the result by Colbeck (1979) and not the result by Schneebeli (1995)?

Reference: MARTIN SCHNEEBELI Biogeochemistry of Seasonally Snow-Covered Catchments (Proceedings of a Boulder Symposium July 1995). IAHSPubl.no. 228,1995. Development and stability of preferential flow paths in a layered snowpack.

- The recurrence of the preferential flow paths that this model is able to simulate was observed to occur in initially dry soil by Wang et al. (2003). However, as other physical processes occur in snow (e.g. snow grain metamorphism) which are not simulated by the model, this section has been removed.

- The citation "Schneebeli (1995)" has been added to the revised manuscript.

Major comments regarding sensitivity study:

- It is not well motivated where the sensitivity study is based on. For example, snow density is varied by 10% , which one can regard as the typical accuracy with which snow density can be determined in

the field. However, the range of densities found in a natural snow cover range from roughly 100 kg/m3 for new snow to 400-500 kg/m3 for old snow and up to 600 kg/m$^3$ for firn. So here, the sensitivity study seems to capture measurement error rather than the range of values found in natural snow covers. On the contrary, the sensitivity study for temperature ranges over 10 degrees. This is the opposite, rather capturing the natural variability found in snow covers than measurement errors.

- The sensitivity analysis has been changed to address this point. The densities and grain sizes now vary to account for a closer approximation of the full range of density and grain size in a natural snow covers.

- A similar comment can be made about the sensitivity study for $\alpha$ and $n$. Where is the choice of a variation of +/- 10% based on? As alpha and n are coupled via rho/d (see Yamaguchi et al. 2012), it is doubtful whether it is an informative result to vary both coefficients separately. I think in the end it is important how much the water retention curve changes. When $n$ is small (1-2), 10 % causes a big change in water retention curve. When n is > 5, the effect is much smaller. Opposite with $\alpha$. When $\alpha$ is large, a 10 % has more influence then when is small. So just modifying alpha and n independently, for just one value of rho/d , is not so informative.

- This sensitivity analysis has been removed. A new sensitivity has been carried on the three different algorithms for $\alpha$ and $n$ from Yamaguchi et al. (2010), Yamaguchi et al. (2012) and Daanen and Nieber (2009) and is described.

- Also grain size is varied over only a very small range. However, grain size has a very important effect on the area that is involved in preferential flow (see for example Katsushima et al. (2013) and Hirashima et al. (2014)). Can the SMPP model reproduce these results?

- The grain size is now varied from 0.1 to 2mm in the sensitivity analysis. The effect of grain size on preferential flow path area is described.

- I can understand the confusion with the irreducible water content. It is true that a similar term is used in the Marsh and Woo papers (1984a,b), although they use the term saturation. It is also true that they used a value of 0.07. However, I do not think that this value is comparable to the role of the residual water content in the water retention curve, where it basically is the lower asymptote of dry conditions. My interpretation of the value used by Marsh and Woo is that the irreducible water saturation is actually the value of Sw (the water saturation) in Equation 1 in the Marsh and Woo (1984a) paper. The saturation is defined between 0 and 1 where 0 is dry snow (or to be precise, snow at residual water content), and 1 is all pores are filled. That means that it should be scaled with the porosity to get the volumetric water content, which would be comparable to the residual water content as used by Yamaguchi et al. (2012). Assume a typical porosity of 0.6 for snow, the irreducible water saturation would translate into a irreducible water content (volumetric) of around 4 % . Furthermore, my

interpretation of their definition of irreducible is more in a bucket type approach, i.e., a typical amount of liquid water that remains in the pores without significant amounts of water flowing, which is not necessarily equal to the dry limit of the van Genuchten water retention curve. In a bucket scheme, typically a value of 4% is used (see for example Wever et al. 2014). Note that field measurements of bulk liquid water content typically ranges from 0.02-0.04 (see Heilig et al. (2015), a reference that deserves citation in this manuscript). Although I also think that the residual water content as used in the water retention curve is likely grain shape and/or grain size dependent, the range used in the sensitivity study (1-10%) doesn't seems to be realistic, given the observational evidence in literature. This is again an example where the choice of range for the sensitivity study is not well motivated, and is actually much larger than for snow density, given the typical range for these properties you will find in nature. As the reference list provided by the authors is not clear at this point, the 1984a paper is for me: Wetting Front Advance and Freezing of Meltwater Within a Snow Cover 1. Observations in the Canadian.

Reference: Heilig, A., C. Mitterer, L. Schmid, N. Wever, J. Schweizer, H.-P. Marshall, and O. Eisen (2015), Seasonal and diurnal cycles of liquid water in snow - Measurements and modeling, J. Geophys. Res. Earth Surf., 120 (10), 2139-2154, doi: 10.1002/2015JF003593.

- There was clearly a misunderstanding between the irreducible water content use in this paper and the one specified in Marsh and Woo (1984a,b). It is also clear that the sensitivity analysis on the irreducible water content was out of bound, therefore, it has been bounded more realistically in the revised manuscript.

- The reference Marsh and Woo (1984a) has been made more clear.

- The research by Heilig et al. (2015) is cited in the revised manuscript.

Minor comments:

- Figure 3 is not really informative, as it is not at all clear if the change in runoff with or without preferential flow has any correspondence with reality. Moreover, the choice to only show the result for density is somewhat arbitrary.

- Figure 3 is now removed from the manuscript with essential information included in a table. The explanation for the oscillation observed in Figure 3 (matrix flow feeding preferential flow paths) has been kept in the revised manuscript as it potentially occurs in natural snow. This phenomenon was also observed while modeling water infiltration into initially dry soil (Jury et al., 2003).

Reference:

Jury, W., Wang, Z., and Tuli, A. : A conceptual model of unstable flow in unsaturated soil during redistribution, Vadose Zone Journal, 2, 61–67, 2003.

- Figure 7: the 4th column of graphs, the title is suddenly expressing the time in seconds, not in hours/minutes.

- • Corrected.

- P3L17: "A melting snow cover can be considered a moving boundary"

- • Corrected.

- P2L28. Should this not read "the minimum suction"? I guess it depends on how positive/negative suction is defined?

- • Corrected.

- Eq 3 is not Richards equation, but just mass conservation. Richards equation is combining mass conservation with Darcy-Buckingham's law. Eq. 3 is valid under many more definitions of the flux q, of which Richards equation is a special category.

- • Corrected.

- Note that Darcy's law is basically the formulation for saturated flow, where the Darcy-Buckingham's law is valid for variable saturation, by introducing a water contents dependence on the hydraulic conductivity.

- • Corrected.

- P4: to be precise: Calonne et al. (2012) developed a relationship for permeability, which can be translated in saturated hydraulic conductivity.

- • Corrected.

- P4, L20: "is" -> "describes"

- • Corrected.

- P4: Eq 6 should be placed after L20-21.

- • Corrected.

- P4L25-26 should move to another place, as first the Equations need to be introduced.

- • Corrected.

- P5L13-14: It seems that here, dry snow is defined when volumetric LWC is below residual water content, and wet when it is above. Yamaguchi's formulation (i.e., the van Genuchten water retention curve) as far as I know is not applicable at all when LWC is below residual water content. So I don't understand this sentence.

- • This sentence has been clarified.

- P6L15: "optimum grid size". Please provide the value for optimum grid size here. I'm also confused

why there is no mention about the time steps? Convergence is often determined by both the time step and the grid cell size. Maybe also mention on what type of computer the simulations are run, and how much CPU time is needed for certain simulations, to give the reader an idea of the computational requirements of the model.

- The value of the optimum grid size varies with the model application. The values are specified in Table 1 of the manuscript.

- It is specified in the manuscript (Section 3) that the time step is adaptive and is calculated to meet the CFL condition.

- The type of CPU that was used and time needed for the simulations have been added in Section 3 of the revised manuscript.

- Section 3.1 is confusing. It sounds as if the snowpack could be considered as being on a slope, where the left-hand side is upslope, and the right hand side is downslope (thus the specific choice of boundary conditions), but this is not explicitly explained and is a bit a puzzle right now.

- The snowpack is level and the side boundary conditions are now set to zero flow. See reply to comment below "P6L27".

- P6L26: Actually, free drainage boundary conditions are a type of Neumann boundary conditions.

- Corrected.

- P6L27: I'm confused about the slope angle. When I'm correct, no result is shown that depends on slope angle? All the results seem to be for a flat snowpack. Furthermore, Eq. 4 is only valid for flat conditions, or when the snowpack is considered vertically (which makes the description of the boundary conditions a bit more complicated). Often in snowpack simulations, the snowpack is considered slope-perpendicular, in which case Eq. 4 needs a modification for the slope angle. Of course, it all depends on definitions of for example the z-coordinate. In any case, the manuscript is confusing at this point and some more clarification is needed.

- The model was developed to be used on both flat and sloped grounds. However, only applications on flat terrain were shown in this manuscript. A demonstration of the model function on a slope is now shown.

- The issue which is addressed on P13L25 seems to be linked to the numerical scheme, but is also not addressed in the appropriate section. So now this point comes out of the blue in the conclusions.

- This issue is now addressed in Section 3.2 as a model approximation.

References:

- Please provide DOIs, should be standard nowadays!

- Added.

- de Rooji: should read de Rooij.

- Corrected.

- The difference between Marsh and Woo, 1984a and b is not made in the reference list. Which one is which?

- Corrected.

- The paper describing SNOWPACK, part II, is having the wrong author list.

- Corrected.

- At least one reference is missing, which is cited in the text (Wever et al. 2014).

- Corrected.